# The Aryl Hydrocarbon Receptor Regulates Invasiveness and Motility in Acute Myeloid Leukemia Cells through Expressional Regulation of Non-Muscle Myosin Heavy Chain IIA

**DOI:** 10.3390/ijms25158147

**Published:** 2024-07-26

**Authors:** Fengjiao Chang, Lele Wang, Youngjoon Kim, Minkyoung Kim, Sunwoo Lee, Sang-woo Lee

**Affiliations:** 1School of Nursing, Shaanxi University of Chinese Medicine, Xianyang 712046, China; cfj611@sntcm.edu.cn; 2Department of Physiology, School of Dentistry and Dental Research Institute, Seoul National University, Seoul 110460, Republic of Korea; wanglele@snu.ac.kr (L.W.); kimyoungjoon@snu.ac.kr (Y.K.); gloriakim@snu.ac.kr (M.K.); sunwoo0458@snu.ac.kr (S.L.); 3Center for Nanoparticle Research, Institute for Basic Science (IBS), Seoul 08826, Republic of Korea

**Keywords:** acute myeloid leukemia, aryl hydrocarbon receptor, extramedullary infiltration, non-muscle myosin heavy chain IIA, chemokinesis, invasion

## Abstract

Acute myeloid leukemia (AML) is the most prevalent type of hematopoietic malignancy. Despite recent therapeutic advancements, the high relapse rate associated with extramedullary involvement remains a challenging issue. Moreover, therapeutic targets that regulate the extramedullary infiltration of AML cells are still not fully elucidated. The Aryl Hydrocarbon Receptor (AHR) is known to influence the progression and migration of solid tumors; however, its role in AML is largely unknown. This study explored the roles of AHR in the invasion and migration of AML cells. We found that suppressed expression of AHR target genes correlated with an elevated relapse rate in AML. Treatment with an AHR agonist on patient-derived AML cells significantly decreased genes associated with leukocyte trans-endothelial migration, cell adhesion, and regulation of the actin cytoskeleton. These results were further confirmed in THP-1 and U937 AML cell lines using AHR agonists (TCDD and FICZ) and inhibitors (SR1 and CH-223191). Treatment with AHR agonists significantly reduced Matrigel invasion, while inhibitors enhanced it, regardless of the Matrigel’s stiffness. AHR agonists significantly reduced the migration rate and chemokinesis of both cell lines, but AHR inhibitors enhanced them. Finally, we found that the activity of AHR and the expression of NMIIA are negatively correlated. These findings suggest that AHR activity regulates the invasiveness and motility of AML cells, making AHR a potential therapeutic target for preventing extramedullary infiltration in AML.

## 1. Introduction

Acute myeloid leukemia (AML) is a highly aggressive cancer of the hematopoietic system and represents the most prevalent form of acute leukemia in adults, making up about 80% of cases. The incidence of AML rises with age, and the 5-year survival rate for patients younger than 60 is approximately 40%, the relapse rate can reach as high as 80%. Elderly patients fare worse, with approximately 70% of those over 65 dying within a year of diagnosis [1]. Recent advances in AML research, such as the development of small molecule targeted therapies (e.g., FLT3 inhibitors, IDH1/2 inhibitors) and immunotherapies (e.g., antibodies against CD33 and CAR-T therapy), have significantly altered clinical treatment paradigms and improved patient outcomes [2,3]. Despite these advancements, high relapse rates and low overall survival remain pressing issues, underscoring the urgent need to further understand AML pathogenesis and develop effective treatment strategies.

Extramedullary infiltration is a major contributor to leukemia relapse or progression to refractory leukemia [4,5]. Common sites of extramedullary infiltration in AML include lymph nodes, liver, spleen, skin, gums, and the central nervous system [6]. Advanced imaging techniques reveal that about 22% of newly diagnosed AML patients experience extramedullary infiltration, with rates rising above 60% following chemotherapy [7]. The mechanisms underlying this infiltration, involving complex processes regulated by multiple factors including chemokines and dynamic remodeling of the extracellular matrix, are not fully understood. The cytoskeletal network, particularly the interaction between non-muscle myosin II (NMII) and F-actin, plays a critical role in regulating cell morphology and migration during this process [8]. Our previous studies have established NMII’s essential role in AML cell migration [9,10], and we have shown that NMIIA is crucial for central infiltration in acute lymphocytic leukemia [11]. Identifying the key upstream regulators of NMIIA in leukemic cells remains a critical research gap.

The Aryl Hydrocarbon Receptor (AHR), a ligand-activated transcription factor originally identified as a mediator of toxic responses to environmental pollutants like 2,3,7,8-tetrachlorodibenzo-p-dioxin (TCDD) [12], has since been linked to a variety of physiological processes. These include angiogenesis, hematopoiesis, drug and lipid metabolism, cell motility, and immune regulation [13]. AHR’s role in malignancy, particularly in cell motility, is crucial for cancer progression. For instance, AHR activation by the tryptophan metabolite kynurenine promotes malignant migration in human glioma cells [14], while omeprazole inhibits tumor migration in breast cancer via the AHR-CXCR4 pathway [15]. These findings illustrate AHR’s specific ligand- and cell type-dependent roles in tumor metastasis.

Based on these insights, we hypothesized that AHR could be a key regulator of leukemic cell motility and extramedullary infiltration. Our study revealed that AHR activity inversely correlates with leukemic cell invasiveness via NMIIA-mediated alterations in chemokinesis. Activation of AHR reduces invasiveness and motility through downregulation of NMIIA, while its inhibition produces the opposite effect. These findings suggest that targeting AHR may be a viable strategy for preventing extramedullary invasion and relapse in leukemia patients.

## 2. Results

### 2.1. Aberrant AHR Targeted Gene Expression and Function in AML

We first examined AHR expression between healthy individuals and AML patients, observing no significant differences (Figure 1A). Subsequently, we analyzed the expression of AHR target genes (CYP1B1, THBS1, SERPINB2) in patients experiencing relapse compared to those in remission, utilizing data from the Vizome database. The results indicated lower expression levels in the relapse group and higher expression in the remission group (Figure 1B–D). These findings imply that the AHR signaling pathway could contribute to enhancing the prognosis of AML patients.

To comprehensively assess the role of AHR signaling in the pathogenesis of AML, we analyzed RNA-Seq data from primary AML specimens treated with the AHR-modulating agent 6-bromoindirubin-3′oxime (BIO, an AHR agonist), sourced from the GEO datasets (GSE48843). This analysis identified 13 upregulated and 12 downregulated genes (|log2FC| > 1.5, *p* < 0.05) post treatment. The top three upregulated genes, as depicted in the volcano plot, were CYP1B1, THBS1, and SERPINB2 (Figure 1E). Further, we performed gene set enrichment analysis (GSEA) to delineate critical signaling pathways impacted by BIO versus DMSO treatment in the AML datasets. This revealed significant associations of AHR activation with various biological processes related to cell movement, including pathways such as “Leukocyte trans-endothelial migration”, “Cell adhesion molecules”, and “Regulation of actin cytoskeleton” (Figure 1F). Specifically, “Leukocyte trans-endothelial migration” and “Regulation of actin cytoskeleton pathways” were notably enriched in the BIO-treated group according to the MSigDB Collection (C2.cp.kegg.v2022.1.Hs.symbols.gmt), as evidenced by their normalized enrichment scores (NESs) and adjusted *p*-values < 0.05 (Figure 1G,H).

### 2.2. Functional Expression of AHR in THP-1 and U937 Cells

We first confirmed the functional expression of AHR in three different AML cell lines: THP-1, U937, and HL-60. Both THP-1 and HL-60 cells showed similar protein-level expression of AHR, whereas U937 cells showed significantly higher expression (Figure 2A). To investigate the effects of AHR activity on the invasive properties of AML across a broader range of baseline AHR expression, we selected THP-1 and U937 cells for further experiments. To verify that AHR is functional in these cells, we observed nuclear translocation of AHR 12 h after treatment with TCDD, an AHR agonist, and SR1, an AHR inhibitor. Western blot analysis of the nuclear/cytoplasm fractions from both THP-1 and U937 cells demonstrated significant nuclear translocation of AHR following TCDD treatment (Figure 2B). Immunofluorescence studies also confirmed that TCDD induced nuclear translocation of AHR, while SR1 treatment resulted in a predominance of cytoplasmic AHR (Figure 2C). Given that activated AHR translocates to the nucleus to function as a transcription factor, initiating transcription of target genes such as CYP1B1, we treated the cells with two AHR agonists (FICZ and TCDD) and two inhibitors (SR1 and CH223191) and then measured the mRNA-level expression of CYP1B1 at 6 and 12 h post treatment. After treating THP-1 and U937 cells with FICZ or TCDD, mRNA expressions of CYP1B1 significantly increased at 6 and 12 h post treatment in both cell lines, confirming that the AHR was functional (Figure 2D). Conversely, treatment with SR1 or CH223191 significantly reduced CYP1B1 mRNA levels in both THP-1 and U937 cells at 6 and 12 h post treatment (Figure 2E).

### 2.3. AHR Activity-Dependent Regulation of Invasiveness of THP-1 and U937 Cells

To examine whether AHR activity can regulate ECM invasion, we conducted Matrigel invasion assays on THP-1 and U937 cells treated with either an AHR agonist or antagonist. To simulate various physicochemical properties of the microenvironment, we varied the Matrigel-to-RPMI ratio, creating soft and stiff Matrigel (Figure 3A). Rheological analysis showed that both soft and stiff Matrigel, with ratios of 1:6 and 1:1, respectively, exhibited gel-like properties (G′ > G″) across all frequency ranges (0.1 to 10 Hz) (Figure 3B). The elastic modulus at 1 Hz was 5.94 Pa for soft Matrigel and 26.40 Pa for stiff Matrigel (Figure 3B). The results of the Matrigel invasion assays indicated that treatment with AHR inhibitors (SR1 and CH223191) significantly increased the invasion of both THP-1 and U937 cells compared to the control group, while treatment with the AHR agonist FICZ significantly reduced it (Figure 3C–E). Notably, THP-1 cells readily invaded both soft and stiff Matrigel, whereas U937 cells demonstrated reduced invasion in stiff Matrigel (Figure 3C). In summary, our results suggest that activation of AHR inhibits AML invasion, whereas inhibition of AHR promotes it.

### 2.4. AHR Activity-Dependent Expressional Regulation of MMP-9 in THP-1 and U937 Cells

To elucidate the mechanism behind the inverse relationship between AHR activity and AML invasion, we examined two possible factors: MMP9 expression and chemokinesis. Initially, MMP9 expression was analyzed in THP-1 and U937 cells after treatment with AHR agonists (TCDD and FICZ) or inhibitors (SR1 and CH223191). The qRT-PCR data indicated that the mRNA-level expression of MMP9 remained unchanged after treatment with either AHR agonists or antagonists in both THP-1 and U937 cells (Figure 4A,B).

### 2.5. AHR Activity-Dependent Regulation of Migration and Chemokinesis in THP-1 and U937 Cells

Next, to assess the effect of AHR activity on the chemokinesis of AML cells, we conducted transwell migration assays and 2D migration assays. In the transwell migration assay, THP-1 and U937 cells were treated with AHR agonists (TCDD and FICZ) or inhibitors (SR1 and CH223191) in the upper well, with 1% FBS in the lower well to initiate migration (Figure 5A, upper). For both cell types, relative migration rates were significantly decreased by treatments with FICZ and TCDD, while treatments with SR1 or CH223191 significantly increased migration (Figure 5B). In the 2D migration assay, the trajectories of THP-1 and U937 cells on a fibronectin-coated surface were monitored over 12 h using a live imaging system after treatment with AHR agonists or inhibitors (Figure 5A, bottom). For both cell lines, migration speed was significantly decreased by treatment with FICZ or TCDD, while it was significantly increased by treatment with SR1 or CH223191 compared to their respective control groups (Figure 5C,D). These results suggest that the AHR activity-dependent increase or decrease in the invasion of THP-1 and U937 cells is due to alterations in chemokinesis.

### 2.6. AHR Activity-Dependent Expressional Regulation of NMIIA in THP-1 and U937 Cells

NMIIA (encoded by the MYH-9 gene) is known to play crucial roles in chemokinesis and cell migration by regulating actin retrograde flow [16] and it also impacts the prognosis of AML due to its influence on extramedullary involvement [17]. To investigate whether AHR activity-induced changes in chemokinesis in THP-1 and U937 cells are linked to the regulation of NMIIA expression, we monitored the mRNA and protein levels of NMIIA at 6 and 12 h post treatment with AHR agonists (FICZ and TCDD) or inhibitors (SR1 and CH223191). In THP-1 cells, treatment with FICZ or TCDD resulted in a decrease in MYH-9 gene expression by 63% and 41%, respectively, compared to control groups at 6 h post treatment (Figure 6A, upper panel). There was no further significant decrease at 12 h (Figure 6A, upper panel). Conversely, treatment with SR1 or CH223191 increased MYH-9 gene expression by 56% and 295%, respectively, at 12 h post treatment (Figure 6A, lower panel), with no significant increase at 6 h (Figure 6A, lower panel). Similar patterns were observed in U937 cells but with greater magnitude. Treatment with FICZ resulted in a 97% and 96% reduction in MYH-9 gene expression at 6 and 12 h post treatment, respectively (Figure 6B, upper panel). TCDD treatment significantly decreased MYH-9 gene expression by 60% and 85% at the respective time points (Figure 6B, upper panel). Meanwhile, treatment with SR1 induced increases in MYH-9 gene expression by 556% and 523% at 6 and 12 h post treatment, respectively (Figure 6B, lower panel). Similarly, treatment with CH223191 increased MYH-9 expression by 560% and 1130% at these time points, respectively (Figure 6B, lower panel).

Western blot data also mirrored the qRT-PCR results. In both THP-1 and U937 cells, treatment with FICZ or TCDD decreased the protein-level expression of NMIIA. Conversely, treatment with SR1 or CH223191 progressively increased NMIIA expression in a time-dependent manner (Figure 6C,D).

To confirm that our hypothesis can be generalizable, we repeated several key experiments on HL-60 cells using the AHR agonist (FICZ) and antagonist (SR1). HL-60 cells responded well to FICZ and SR1 by significantly increasing and decreasing CYP1B1 expressions, respectively (Appendix A). Similar to the results from THP-1 and U937 cells, HL-60 cells also showed an increased transwell migration rate and NMIIA expression level when treated with FICZ, while the migration rate and NMIIA expression level decreased when treated with SR1, with no change in MMP9 expression (Appendix A).

## 3. Discussion

The AHR signaling pathway plays an important role in hematologic malignancies. Recent studies have found that silencing AHR promotes apoptosis in chronic lymphocytic leukemia (CLL) cells and synergistically enhances the cytotoxicity of the chemotherapy drug venetoclax [18]. Another study published in *Cell* indicated that activation of the AHR pathway also accelerates the metastatic infiltration of CLL cells in the spleen and lymph nodes [19]. Interestingly, the role of AHR in myeloid leukemia is quite the opposite. Gentil et al. found that AHR is underexpressed in patients with chronic myeloid leukemia (CML), and activating AHR can effectively inhibit leukemia cell proliferation [20]. Ly et al. revealed that AHR target genes are downregulated in AML patients, and activating this pathway can serve as a core signal to promote AML cell differentiation and inhibit cell self-renewal [21]. Our results demonstrated that activation of the AHR inhibited the invasiveness of AML cells through the downregulation of NMIIA. Typically, the canonical AHR activation pathway is known to facilitate transcriptional activation of target genes. However, our findings, which show AHR-mediated suppression of the MYH-9 gene, suggest that this effect might be a secondary outcome of the canonical AHR pathway. For example, AHR can induce transcription of micro RNAs (miRNAs) to downregulate transcription of target mRNAs. Therefore, we suspect that AHR activation transcribed a pool of miRNA targeting the MYH-9 mRNA. There is a report that AHR-mediated over expression of miR-212/132 cluster reduced migration and invasion of breast cancer cells by suppressing mRNA of pro-metastatic transcription factor SOX-4 [22]. Further investigation of the details of this mechanism is required.

Therefore, further discovery of effective ligands that can regulate AHR signaling to affect AML extramedullary infiltration has significant clinical value in controlling AML relapse. Although various AHR agonists, including TCDD, FICZ, and Bap, have demonstrated therapeutic potential in in vitro models, their use can lead to severe complications such as liver fibrosis [23], embryo mortality [24] and carcinogenesis [25]. Currently, only Tapinarof topical cream, an AHR agonist, has reached phase 3 trials, and it is being tested for limited indications such as psoriasis and atopic dermatitis [26]. Given these concerns, there is an urgent need to identify AHR agonists that are safe and effective for systemic use. Recently, naturally derived compounds like indole-3-carbinol, resveratrol, curcumin, and polyphenols have been proposed as alternatives to conventional AHR agonists, exhibiting relatively lower toxicity than exogenous synthetic AHR ligands such as TCDD [27]. However, the use of an AHR agonist or antagonist in clinical settings should be conducted with caution since the level of AHR activation has to be precisely regulated in physiological conditions. For an example, I3C, an AHR agonist, can reduce inflammatory responses in the gut, but it simultaneously accelerates formation of colonic lesions, which may become a tumor over the long-term [28]. Therefore, it is extremely important to consider the local concentration, potency, duration of action, and pharmacokinetics of AHR regulators in order to use them in clinical settings. Another challenge in the clinical application of AHR agonists is their low solubility, which often requires the use of toxic organic solvents or complex drug delivery systems.

The process of leukemia extramedullary infiltration involves leukemia cells being released from the bone marrow and disseminating into the peripheral blood circulation. Subsequently, under the influence of chemokines and integrins, they adhere to the vascular endothelium and crawl along the endothelial cell barrier. When they find a permissible location, leukemia cells dynamically reshape their morphology between endothelial cells, thereby crossing from within the vessel to outside, and then migrate and invade extramedullary tissues and organs conducive to their growth and proliferation [29]. We have simulated the crawling, endothelial barrier escaping, and tissue invading process using a 2D migration assay, a transwell migration assay, and Matrigel invasion assay. It is noteworthy that U937 cells, which express AHR at high levels, exhibit lower invasiveness compared to THP-1 cells, which have lower AHR expression. This difference suggests that the basal expression level or activity of AHR might be predictive of extramedullary infiltration in AML. However, no statistical data directly linking AHR expression levels with relapse rates have been found. Instead, what we have found in the database is that the expression level of downstream AHR genes—which could reflect either the level of AHR expression or its activity—is inversely correlated with the relapse rate. To further verify it, genetic overexpression or a knock-out study will be required. The difference in the basal expression levels of AHR in THP-1 and U937 cells is possibly due to the different tissue origin and maturation state. U937 cells are matured tissue-origin AML cell lines, while THP-1 cells are from the blood and are less matured [30]. Considering that both THP-1 cells and HL-60 cells are from the blood and show similar basal expression levels of AHR, the origin of the cells seems crucial. MMP-9 is another factor contributing to cancer cell migration and metastasis. It has been reported that AHR activation in gastric and prostate cancer cells induces MMP-9 upregulation [31,32]. Leukemia cells can also secrete MMP-9, which is known to facilitate progression and invasion of AML [33]. However, the mRNA level of MMP-9 has not been changed by the activation/inactivation of AHR in THP-1 and U937 cells, which is different from that in solid cancer. Another limitation of this study is that only MMP9 and NMIIA were examined as downstream targets of AHR. Expressional changes in chemokine receptors such as CXCR4 or CCR5 can also affect migration and invasion of AML cells [34]. Therefore, it should also be further examined whether activation or inhibition of AHR can alter expression levels of those genes.

## 4. Materials and Methods

### 4.1. Data Sources

Vizome (http://vizome.org/aml2 (accessed on 16 January 2024)) is the largest public database for AML and was used to analyze the AHR- and AHR-targeted gene expression levels between AML and normal controls. The AML expression profiling dataset GSE48843, containing expression data for primary AHL specimens treated with AHR-modulating agents, was obtained from GEO (https://www.ncbi.nlm.nih.gov/geo (accessed on 9 March 2024)).

### 4.2. Bioinformatics Analysis

Hierarchical clustering based on RNA expression levels between DMSO- and BIO-treated mRNA expression groups was performed and visualized using volcano plots with the ggplot2 package in the R environment, utilizing data from GEO. GSEA software (version 4.3.2, Cambridge, MA, USA) was used to elucidate the functional and pathway differences. The gene set was permutated 1000 times for each analysis. The normalized enrichment score (NES), nominal *p* value (*p*), and false discovery rate q value (FDR q) were selected to classify enriched signal pathways. Gene sets with |NES| > 1, NOM *p* < 0.05, and FDR q < 0.25 were regarded as significantly enriched.

### 4.3. Cells

After purchasing THP-1, HL-60, and U937 from the Seoul National University Cell Bank, we cultured them in RPMI-1640 medium (Hyclone, Washington, DC, USA, SH30027.01) including 10% FBS (Hyclone, SH30084.03), 1% sodium pyruvate (Gibco, 11360-070), and 1% penicillin (Gibco, Waltham, MA, USA, 15140-122) at 37 °C in a 5% CO_2_ incubator. The cells were used during their logarithmic growth phase for the experiment. FICZ (6-formylindolo [3,2-b] carbazole; 100 nM), TCDD (D-404N, AccuStandard, New Haven, CT, USA; 100 nM), StemRegenin 1 (SR1) (C7710-1, Cellagen Technology; 1 µM), and CH223191 (3858, Tocris Bioscience, Bristol, UK; 1 µM) were added to the cells. 

### 4.4. Boyden Chamber Migration Assay

Transwell assays were conducted to assess the chemokinetic effects of FICZ, TCDD, SR1 and CH223191 on cells. THP-1, HL-60, and U937 cells were maintained in the upper chamber of a transwell plate with 8µm diameter pores and filled with serum-free medium. The relevant reagents were appropriately diluted and added to the upper chamber. Subsequently, the cells were incubated for a predetermined period to facilitate chemokinesis studies. After completion of the experiment, migrated cells in the lower layer were collected and quantitatively analyzed using hemocytometry along with appropriate statistical methods.

### 4.5. Immunofluorescence Staining

THP1 and U937 cells were maintained in 96-well plates (0611129L2L, Matrical Bioscience, Spokane, WA, USA) that had been coated with fibronectin (F0895, Sigma, St. Louis, MO, USA) before the cells were cultured. After scheduled treatment, the cells were washed twice with PBS followed by fixation in 4% paraformaldehyde for 20 min at room temperature. Subsequently, the cells were washed 3 times with 0.1% PBST (PBS added to 0.1% Tween20) and permeabilized with 0.1% Triton X-100 for 15 min at room temperature. Following an hour of blocking solution incubation, the cells were incubated overnight at 4 °C with primary antibodies. The following were the primary antibodies: Mouse AHR antibody (sc-133088). Afterward, the cells were incubated with Alexa Fluor 488 rabbit anti-mouse IgG for 2 h at room temperature. The nuclei and cytoskeleton were stained with DAPI (Vector Laboratories, Burlingame, CA, USA) and phalloidin (T7471, Invitrogen, Waltham, MA, USA), respectively. Images were acquired using a laser scanning confocal microscope (LSM 700/980, Carl Zeiss, Jena, Germany).

### 4.6. Western Blot

THP-1, HL-60, and U937 cells were washed twice with cold PBS and subsequently lysed using RIPA lysis buffer (R4200-010, GenDEPOT, Katy, TX, USA) containing a protease inhibitor cocktail and phosphate inhibitor (Thermo Scientific, Waltham, MA, USA, 1861280). The lysates were then kept on ice for 30 min. After sonication, the samples were centrifuged at 12,000 rpm for 30 min at 4 °C. The supernatant was collected, and its protein concentration was determined using a BCA protein assay kit (23227, Pierce, Appleton, WI, USA), following the manufacturer’s instructions. For the nuclear translocation assay, a nuclear extraction kit (ab113474, Abcam, Cambridge, UK) was employed.

Subsequently, the protein mixtures were separated using SDS-PAGE gels and then transferred onto membranes with an iBLOT 2 Dry Blotting system (IB21001, Thermo Scientific). To prevent non-specific binding on the membranes, either 5% non-fat milk or 5% bovine serum albumin (BSA) was applied. Primary antibodies against NMIIA (909801, Biolegend, San Diego, CA, USA) and AHR (sc-133088) were applied to detect their presence in the samples. Following this, secondary antibody (goat anti-mouse, K0211589, goat anti-rabbit, k0211708, LABISKOMA) were added at room temperature. Visualization of target proteins was achieved using ECL reagents (32106, Thermo Scientific), followed by detection with the Chemidoc XRS+ system (Bio-Rad Laboratories, Hercules, CA, USA).

### 4.7. Reverse Transcriptase-PCR and Real-Time PCR

After harvesting THP-1, HL-60, and U937 cells, RNA extraction was performed using the RNeasy Mini Kit (74140, Qiagen, Germantown, MD, USA) following the manufacturer’s instructions. For cDNA synthesis, 1 ug of purified RNA was reverse transcribed using SuperScript™ ⅣReverse Transcriptase (18090-050, Invitrogen). Quantitative real-time PCR was conducted on a 7500 Real-Time PCR System (Applied Biosystems, Waltham, MA, USA) using SYBR PCR master mix (4309155, Applied Biosystems). The primer sequences used were as follows:

GAPDH: forward 5′-CGACCACTTTGTCAAGCTCA-3′, and reverse 5′-AGAGTTGTCAGGGCCCTTTT-3′;CYP1B1, forward 5′-ACGTACCGGCCACTATCACT-3′, and reverse 5′-CTCGAGTCTGCACATCAGGA-3′;MYH9: forward 5′-CCCAGAAGAGGAGCAAATGG-3′, and reverse 5′-GTAATCCCGTCCCACCTTGA-3′.CXCR4: forward 5′-CTCCAAGCTGTCACACTCCA-3′, and reverse 5′-GAGTCGATGCTGTCCCAAT-3′.

### 4.8. 2-Dimensional (2-D) Migration Assay

THP-1 and U937 cells (2 × 10^5^/well) were pre-seeded in the incubator overnight. On the following day, the cells were cultured in serum-free media in the incubator for 4 h. Subsequently, the cells were transferred to fibronectin (F0895, Sigma)-coated 96-well plates (0611129L2L, Matrical Bioscience, Spokane, WA, USA) that had been pre-coated with fibronectin for 30 min. Utilizing a JuLi Br Live Cell Analyzer (JULI-BR04, NanoEnTek Inc. Seoul, Korea) equipped with an 8X objective lens, time-lapse images were recorded every 3 min over a span of 12 h. The speed and quantity of cells were analyzed using Image J software 1.54f released at June 2023 (National Institutes of Health, Bethesda, MD, USA).

### 4.9. Invasion Assay

The 24-well transwell chamber was washed twice with 200 µL of serum-free RPMI media before adding the diluted Matrigel (356231, corning, Corning, NY, USA) at a ratio of 1:1 or 1:6 to the upper chamber and incubated overnight in an incubator. Subsequently, logarithmic growth phase cells were rinsed to remove serum and then seeded in the upper chamber at a density of (3 × 10^5^)/mL using serum-free RPMI-1640 medium with a volume of 100 µL. In the bottom, 600 µL of media containing 15% FBS was added and cultured for 24 h. The upper chamber was fixed with 200 µL of 4% paraformaldehyde for 30 min at room temperature and washed twice with PBS. Afterward, staining was performed using a solution of 1% crystal violet for 30 min followed by rinsing several times with distilled water (DDW). Then, the upper chamber was removed and at least five random areas were selected for imaging using Nikon equipment (Tokyo, Japan). The stained cells within each area were counted and subjected to statistical analysis.

### 4.10. Rheological Analysis

A rotational rheometer (Kinexus, Lab+, Netzsch, Germany) was used to measure the rheological properties of soft (Matrigel: RPMI = 1:6) and stiff (Matrigel: RPMI = 1:1) Matrigel. The elastic modulus (G′) and viscous modulus (G″) of the Matrigel were quantified as a function of frequency.

### 4.11. Statistical Analysis

The data analysis was performed using the statistical methods and the results were presented as mean ± S.E.M. *T* tests and analysis of variance (ANOVA) were employed for intergroup comparison, with a significance threshold of *p* < 0.05.

## 5. Conclusions

In summary, we first found a significant correlation between AHR activity and the relapse rate of AML. Analysis of mRNA sequencing data from AML cells treated with AHR agonists revealed that AHR activation reduces gene expression related to migration, cell adhesion, and the cytoskeleton. Using THP-1 and U937 cells, we demonstrated that AHR activity can regulate the invasiveness and chemokinesis of AML cells. Activation of AHR with agonists significantly decreased the invasiveness and chemokinesis of these cells, whereas inhibition of AHR had the opposite effect. Importantly, these changes were mediated by the expressional regulation of NMIIA rather than MMP-9, indicating a putative novel molecular mechanism through which AHR influences leukemic cell behavior.

## Figures and Tables

**Figure 1 ijms-25-08147-f001:**
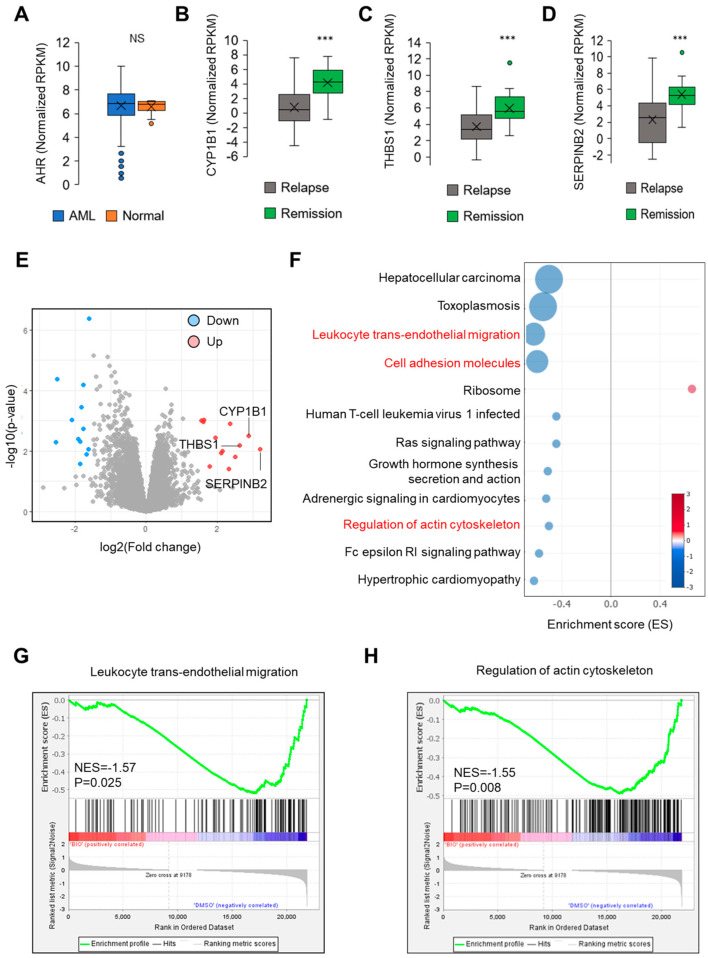
(**A**) Normalized RPKM of AHR expression between normal controls and AML patients in the Vizome database. (**B**–**D**) Normalized RPKM of AHR targeted genes (CYP1B1, THBS1, and SERPINB2, respectively) in the relapse and remission phase of AML from the Vizome database. ***: *p* < 0.001. (**E**) Volcano plot of differentially expressed genes in AML patients treated with DMSO and BIO of the GSE48843 database. The thresholds are |log2FC| > 1.5, *p* < 0.05. (**F**) Top 12 enriched GSEA-KEGG pathways in the DMSO- versus BIO-treated AML groups. Key KEGG pathways are marked in red. (**G**,**H**) Selected biological process associated GSEA pathways in the BIO-treated group. *p* values are labeled in this Figure. NS: non-significant.

**Figure 2 ijms-25-08147-f002:**
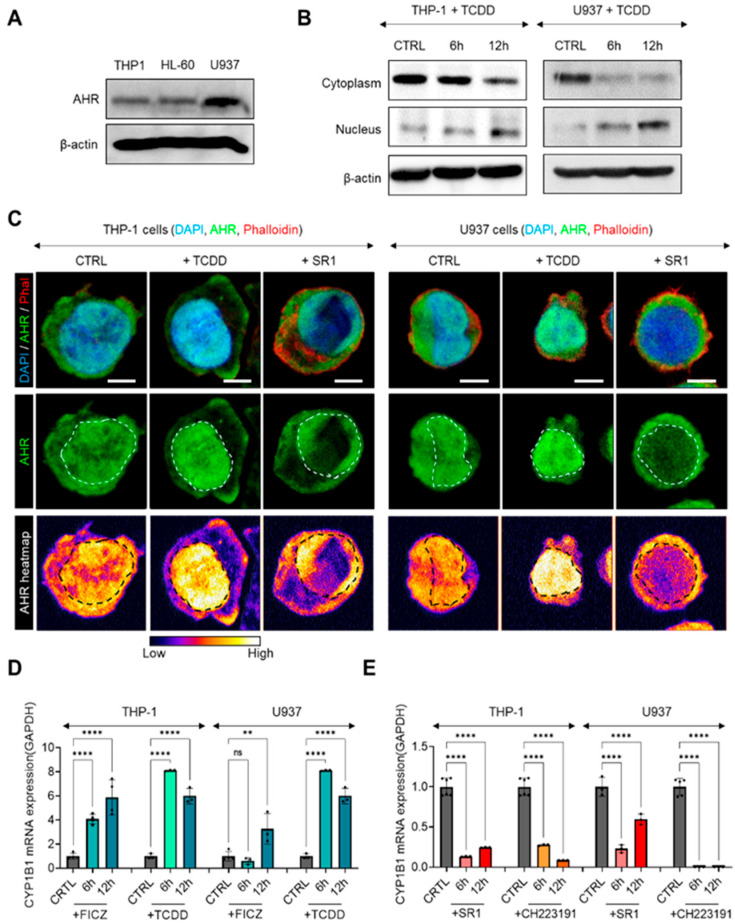
(**A**) Western blot of AHR (100 kDa) and β-actin (42 kDa) expressions in THP-1, HL-60, and U937 cell lines. (**B**) Western blot of nuclear and cytoplasmic protein fraction of AHR (100 kDa) in THP-1 and U937 cells treated with TCDD for 6 and 12 h. β-actin (42 kDa) bands are from cytoplasmic fraction. (**C**) Subcellular localization of AHR in THP-1 and U937 cells after being treated with TCDD or SR1 (DAPI in blue, AHR in green, Phalloidin in red). Relative expression heatmap of AHR is in third row. Locations of nuclei are indicated with white and black dotted lines. Scale bar = 5 μm. (**D**,**E**) Relative mRNA expression levels of the CYP1B1 gene 6 and 12 h after FICZ, TCDD, SR1, and CH223191 treatment. Expression levels are normalized to GAPDH. Data are expressed as average ± s.e.m. One-way ANOVA is performed with Tukey’s multiple comparison tests. Significance is set to ns: non-significant; **: *p* < 0.01; ****: *p* < 0.001. Images in each corresponding group were equally enhanced in PowerPoint to provide better visual clarity.

**Figure 3 ijms-25-08147-f003:**
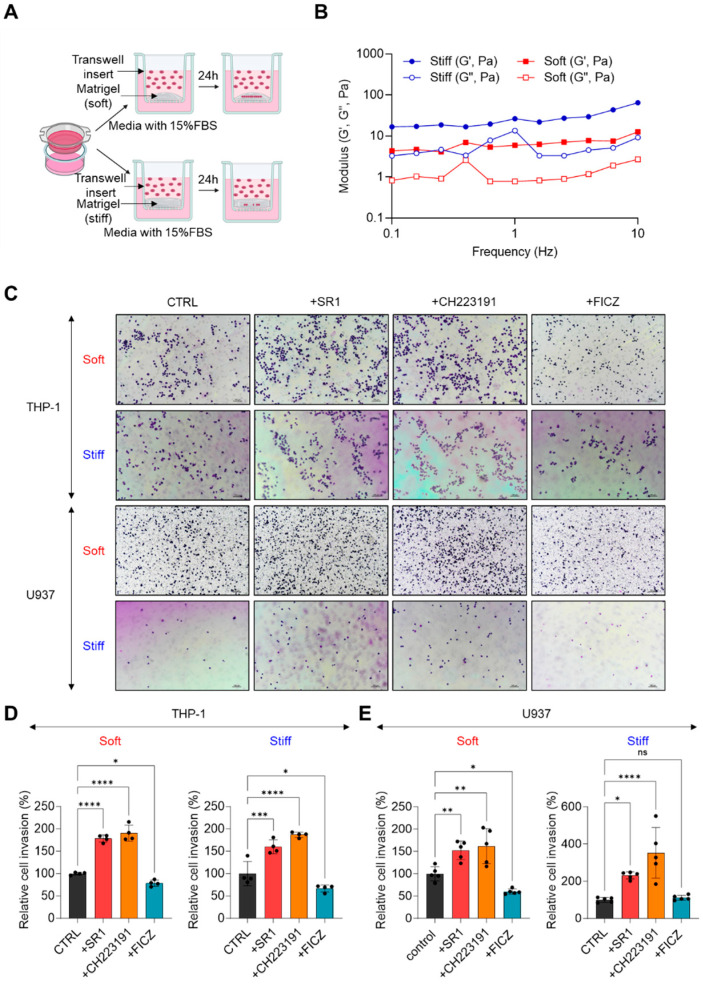
(**A**) Schematic diagram of Matrigel invasion assay of THP-1 and U937 cells. (**B**) Rheological analysis of soft and stiff Matrigel. G′ (elastic modulus) and G′′ (viscous modulus) under frequency sweep mode (0.1–10 Hz). (**C**) Representative images from the Matrigel invasion assay results. THP-1 and U937 cells treated with SR1, CH223191, or FICZ. Cells invaded through Matrigel are stained with crystal violet (purple). Scale bar = 100 μm. (**D**,**E**) The relative quantifications of Matrigel-invaded THP-1 cells and U937 cells. CTRL group as 100%. Data are expressed as average ± s.e.m. One-way ANOVA is performed with Tukey’s multiple comparison tests. Significance is set to ns: non-significant *: *p* < 0.05; **: *p* < 0.01; ***: *p* < 0.005; ****: *p* < 0.001. Images in each corresponding group were equally enhanced in PowerPoint to provide better visual clarity.

**Figure 4 ijms-25-08147-f004:**
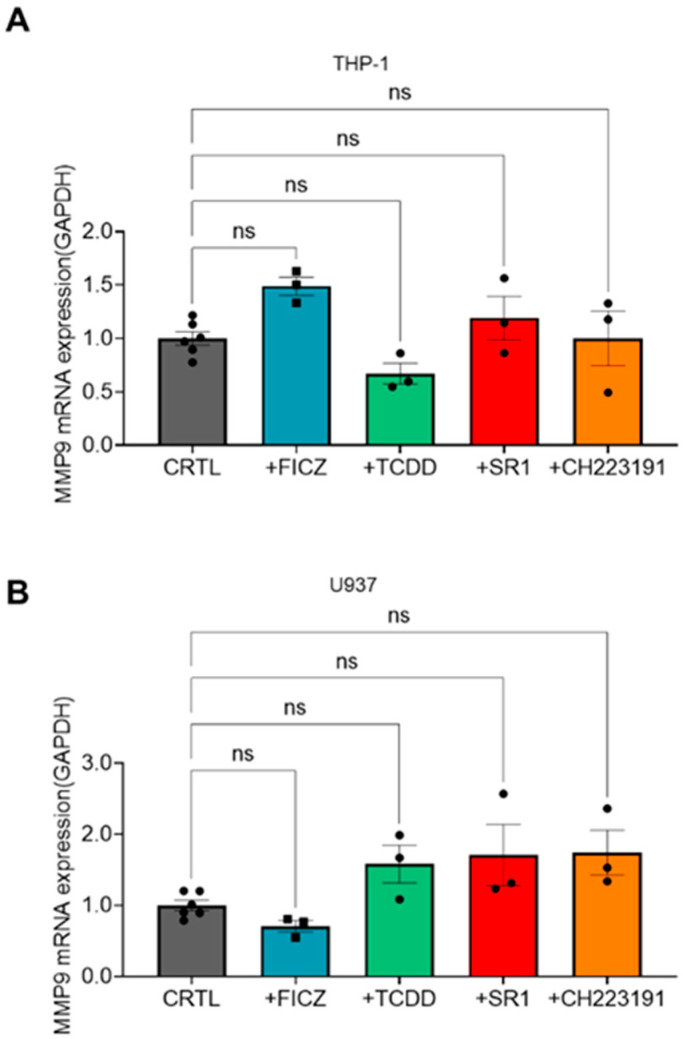
(**A**) Relative mRNA expression levels of MMP-9 gene in THP-1 cells 12 h after FICZ, TCDD, SR1, and CH223191 treatment. Expression levels are normalized to GAPDH. CTRL was set to 1. (**B**) Relative mRNA expression levels of MMP-9 gene in U937 cells 12 h after FICZ, TCDD, SR1, or CH223191 treatment. Expression levels are normalized to GAPDH. CTRL is set to 1. One-way ANOVA is performed with Tukey’s multiple comparison tests. Significance is set to ns: non-significant.

**Figure 5 ijms-25-08147-f005:**
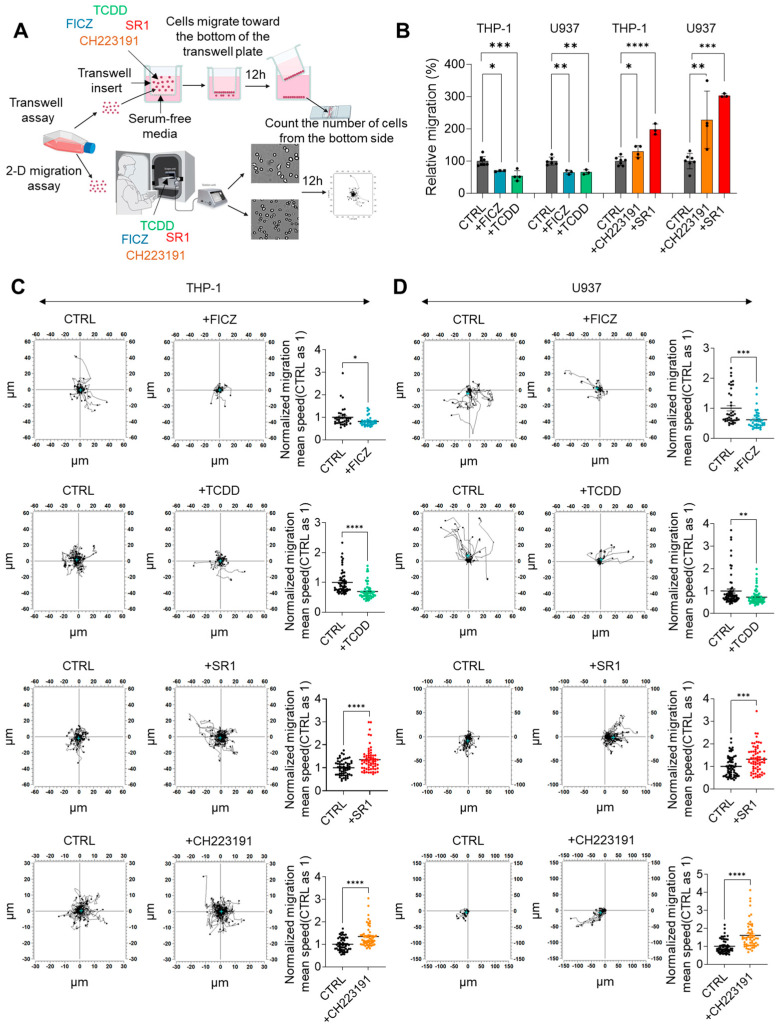
(**A**) Schematic diagram of transwell migration assay and 2-D migration assay of THP-1 and U937 cells. (**B**) Quantification of relative migration of THP-1 and U937 cells treated with FICZ, TCDD, SR1, or CH223191. CTRL is set to 1 in each experimental set. (**C**) Migration trajectory and relatively quantified migration mean speed of THP-1 cells treated with FICZ, TCDD, SR1, or CH223191. CTRL is set to 1 in each experimental set. (**D**) Migration trajectory and relatively quantified migration mean speed of U937 cells treated with FICZ, TCDD, SR1, or CH223191. CTRL is set to 1 in each experimental set. One-way ANOVA is performed with Tukey’s multiple comparison tests for (**B**) and unpaired *t*-test is performed for (**C**,**D**). Significance is set to *: *p* < 0.05; **: *p* < 0.01; ***: *p* < 0.005; ****: *p* < 0.001.

**Figure 6 ijms-25-08147-f006:**
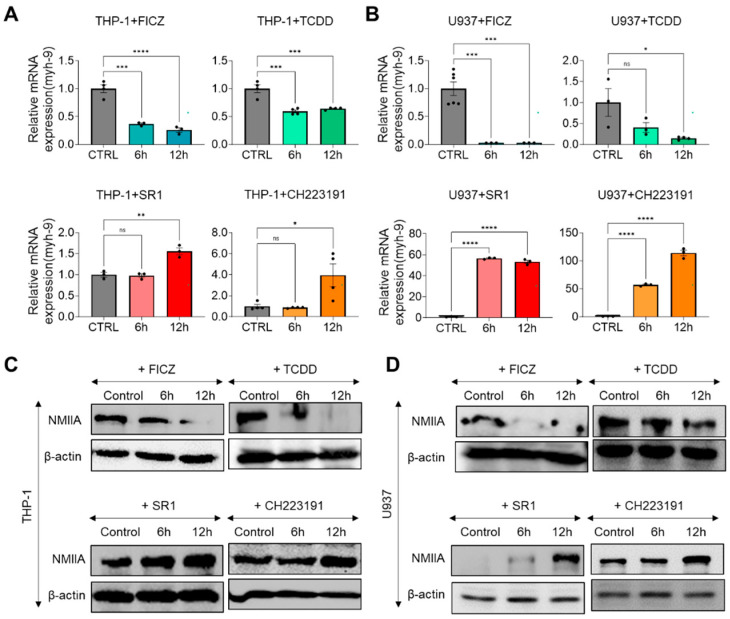
(**A**) Relative mRNA expression levels of *MYH-9* gene in THP-1 cells 6 and 12 h after FICZ, TCDD, SR1, and CH223191 treatment. Expression levels are normalized to GAPDH. CTRL was set to 1. (**B**) Relative mRNA expression levels of *MYH-9* gene in U937 cells 6 and 12 h after FICZ, TCDD, SR1, or CH223191 treatment. Expression levels are normalized to GAPDH. CTRL was set to 1. (**C**) Western blot of NMIIA (200 kDa) and β-actin (42 kDa) expressions in THP-1 cells 6 and 12 h after treatment with FICZ, TCDD, SR1, or CH223191. (**D**) Western blot of NMIIA (200 kDa) and β-actin (42 kDa) expressions in U937 cells 6 and 12 h after treatment with FICZ, TCDD, SR1, or CH223191. One-way ANOVA is performed with Tukey’s multiple comparison tests. Significance is set to ns: non-significant; *: *p* < 0.05; **: *p* < 0.01; ***: *p* < 0.005; ****: *p* < 0.001. Images in each corresponding group were equally enhanced in PowerPoint to provide better visual clarity.

## Data Availability

Data are available from the corresponding author.

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
