# Peer review of "The Aryl Hydrocarbon Receptor Regulates Invasiveness and Motility in Acute Myeloid Leukemia Cells through Expressional Regulation of Non-Muscle Myosin Heavy Chain IIA"

_ijms, 2024, doi:10.3390/ijms25158147_

Round 1

Reviewer 1 Report

Comments and Suggestions for Authors

Authors elucidated the relationship of the AHR with invasiveness and motility in AML. Authors of IJMS are interested in those results. Therefore, I think this manuscript is worthy being published in IJMS.

It is not clear why there is a difference in AHR expression between THP-1 and U937.

Treatment with agonists and inhibitors, the concentrations of agonists and inhibitors were not described. Does dependent manner of agonists and antagonists may be important.

About expression of relative mRNA expression (myh-9), treatment of CH223191 is time dependent, while treatment of SR1 is the same after 6 and 12h. Why treatment of SR1 gave the same results after 6 and 12h?

Author Response

  1. It is not clear why there is a difference in AHR expression between THP-1 and U937.

Response:

We appreciate reviewer’s question. Although we do not have fundamental explanation for the difference in AHR expression between THP-1 and U937 cells, we believe that the difference in basal expression levels of AHR in THP-1 and U937 cells is possibly due to the different tissue origin and maturation state. U937 cells are matured tissue-origin AML cell lines, while THP-1 cells are from blood and less matured [Chanput, Wasaporn, Vera Peters, and Harry Wichers. "THP-1 and U937 Cells." The Impact of Food Bioactives on Health: in vitro and ex vivo models (2015): 147-159.]. Considering that both THP-1 cells and HL-60 cells are from blood and show similar basal expression levels of AHR, origin of cells seems crucial. We discussed this issue in our discussion section.

Revised manuscript: Discussion / Page XX, Line YY

To further verify it, genetic overexpression or knock-out study will be required. The difference in basal expression levels of AHR in THP-1 and U937 cells is possibly due to the different tissue origin and maturation state. U937 cells are matured tissue-origin AML cell lines, while THP-1 cells are from blood and less matured [Chanput, Wasaporn, Vera Peters, and Harry Wichers. "THP-1 and U937 Cells." The Impact of Food Bioactives on Health: in vitro and ex vivo models (2015): 147-159.]. Considering that both THP-1 cells and HL-60 cells are from blood and showing similar basal expression levels of AHR, origin of cells seems crucial. MMP-9 is another factor contributing cancer cell migration and metastasis.

  1. Treatment with agonists and inhibitors, the concentrations of agonists and inhibitors were not described. Does dependent manner of agonists and antagonists may be important.

Response:

Thank you for your valuable insight. For FICZ, the concentration used is 100 nM. For TCDD, it is also 100 nM. The concentration for SR1 is 1µM, and for CH223191, it is 1µM as well. We added this information to our Materials and Methods section 2.3.

We deeply agree that the dependent test of agonist and antagonist is important. However, due to the limited resources and time, we took the optimal concentrations of those reagents already tested in our previous study [Chang, Fengjiao, et al., "Targeting actomyosin contractility suppresses malignant phenotypes of acute myeloid leukemia cells." International Journal of Molecular Sciences 21.10 (2020): 3460.].

Revised manuscript: Materials and Method / Page XX, Line YY

2.3. Cells

After purchasing THP-1, HL-60 and Uwith 10% FBS (Hy-clone, SH30084.03), 1% sodium pyruvate (Gibco,11360-070), and 1% penicillin (Gib-co,15140-122) at 37°C in a 5% CO2 incubator. The cells were used during the logarithmic growth phase for the experiment. FICZ (6-formylindolo [3,2-b] carbazole; 100 nM), TCDD (D-404N, AccuStandard; 100 nM), StemRegenin 1 (SR1) (C7710-1, Cellagen Technology; 1 µM), and CH223191 (3858, Tocris Bioscience; 1 µM) were added to the cells for 6 hours or 12 hours.

  1. About expression of relative mRNA expression (myh-9), treatment of CH223191 is time dependent, while treatment of SR1 is the same after 6 and 12h. Why treatment of SR1 gave the same results after 6 and 12h?

Response:

We appreciate the reviewer’s comment. Although we do not have fundamental answer for this one, the difference in the tendency of time-dependent enhancement of myh-9 gene expression can be attributed to the difference in their potency (IC50). The IC50 of SR1 is approximately 127 nM, while the IC50 of CH223191 is 30 nM, indicating that CH223191 is a more potent inhibitor. Therefore, treatment of U937 cells with SR1 induced approximately a 55-fold increase in myh-9 gene expression at 6 hours post-treatment, which remained the same at 12 hours post-treatment. In contrast, treatment with the highly potent CH223191 induced about a 50-fold increase in myh-9 gene expression at 6 hours, which continuously increased to approximately a 110-fold increase at 12 hours. These results indicate a continuous accumulation of myh-9 mRNA in the cells with more effective and prolonged activation of AHR.

Revised manuscript: We have not revised regarding this issue.

Reviewer 2 Report

Comments and Suggestions for Authors

The study explores the role of the Aryl Hydrocarbon Receptor (AHR) in regulating invasiveness and motility in Acute Myeloid Leukemia (AML) cells. Key findings include:

1.     A significant correlation between AHR activity and relapse rate in AML.

2.     AHR activation reduces gene expression related to migration, cell adhesion, and cytoskeleton in AML cells.

3.     Activation of AHR with agonists decreases invasiveness and chemokinesis, while inhibition increases these traits.

4.     These effects are mediated by NMIIA (non-muscle myosin heavy chain IIA) rather than MMP-9, suggesting a novel molecular mechanismConclusionsThe study concludes that AHR activity can regulate the invasiveness and motility of AML cells, providing potential therapeutic targets for reducing AML relapse rates by modulating AHR activity and NMIIA expression​​.

The writing is clear and concise, effectively communicating complex biological processes and experimental results. The structured format with clear subheadings helps in understanding the methodology and findings. However, some sentences are dense and could be simplified for better readability.

Major concerns:

1.     Specificity of AHR effects: The study suggests that AHR's effects are mediated by NMIIA rather than MMP-9. Further exploration is needed to confirm this pathway and rule out other potential mediators.

2.     Therapeutic Application: While the findings are promising, the study does not address potential side effects or the feasibility of targeting AHR in clinical settings.

3.     Sample Size: The study's conclusions are based on experiments with specific AML cell lines (THP-1 and U937). Broader studies involving diverse AML samples would strengthen the generalizability of the results.

Minor questions, which should at least be discussed:

1.     Mechanism of AHR Regulation: What are the detailed molecular pathways through which AHR regulates NMIIA and other cytoskeletal components in AML cells?

2.     Clinical Feasibility: Can AHR agonists be safely and effectively used in patients, and what would be the potential side effects?

3.     Broader Implications: How does AHR activity interact with other signaling pathways in AML and other types of leukemia?

4.     Long-term Effects: What are the long-term outcomes of modulating AHR activity in AML treatment? Will it lead to sustained remission or just a temporary reduction in invasiveness?

Comments on the Quality of English Language

The writing is clear and concise, effectively communicating complex biological processes and experimental results. The structured format with clear subheadings helps in understanding the methodology and findings. However, some sentences are dense and could be simplified for better readability.

Author Response

Reviewer#1

Comments to the Author

The writing is clear and concise, effectively communicating complex biological processes and experimental results. The structured format with clear subheadings helps in understanding the methodology and findings. However, some sentences are dense and could be simplified for better readability.

Major concerns:

  1. Specificity of AHR effects: The study suggests that AHR's effects are mediated by NMIIA rather than MMP-9. Further exploration is needed to confirm this pathway and rule out other potential mediators.
  2. Therapeutic Application: While the findings are promising, the study does not address potential side effects or the feasibility of targeting AHR in clinical settings.
  3. Sample Size: The study's conclusions are based on experiments with specific AML cell lines (THP-1 and U937). Broader studies involving diverse AML samples would strengthen the generalizability of the results.

Minor questions, which should at least be discussed:

  1. Mechanism of AHR Regulation: What are the detailed molecular pathways through which AHR regulates NMIIA and other cytoskeletal components in AML cells?
  2. Clinical Feasibility: Can AHR agonists be safely and effectively used in patients, and what would be the potential side effects?
  3. Broader Implications: How does AHR activity interact with other signaling pathways in AML and other types of leukemia?
  4. Long-term Effects: What are the long-term outcomes of modulating AHR activity in AML treatment? Will it lead to sustained remission or just a temporary reduction in invasiveness?

Major concerns:

  1. Specificity of AHR effects: The study suggests that AHR's effects are mediated by NMIIA rather than MMP-9. Further exploration is needed to confirm this pathway and rule out other potential mediators.

Response:

We would like to thank you for taking the time and effort to provide comments on the manuscript. As the reviewer suggested, we added this in our discussion part as the limitation of this study.

Revised manuscript: Discussion / Page XX, Line YY

“Another limitation of this study is that only MMP9 and NMIIA were examined as down-stream targets of AHR. Expressional changes of chemokine receptors such as CXCR4 or CCR5 can also affect the migration and invasion of AML cells [ref]. Therefore, it should also be further examined whether activation or inhibition of AHR can alter expression levels of those genes.”

  1. Therapeutic Application: While the findings are promising, the study does not address potential side effects or the feasibility of targeting AHR in clinical settings.

Response:

We thank for the reviewer’s comment. As the reviewer pointed out, AHR regulators are double-sided sword which can exert both positive and negative effects. For example, I3C, an AHR agonist, can reduce inflammatory response in gut, but it simultaneously accelerates formation of colonic lesion, which may become a tumor in long-term manner [Chen, Yue, et al., "Modulating AHR function offers exciting therapeutic potential in gut immunity and inflammation." Cell & Bioscience 13.1 (2023): 85.]. Therefore, it is extremely important to consider local concentration, potency, duration of action, and pharmacokinetics of AHR regulators in order to use them in clinical settings. We added aforementioned statements in our discussion section.

Revised manuscript: Discussion / Page XX, Line YY

Given these concerns, there is an urgent need to identify AHR agonists that are safe and effective for systemic use. Recently, naturally derived compounds like indole-3-carbinol, resveratrol, curcumin, and polyphenols have been proposed as alternatives to conventional AHR agonists, exhibiting relatively lower toxicity than exogenous synthetic AHR ligands such as TCDD [27]. However, use of AHR agonist or antagonist in clinical setting should be cautious since the level of AHR activation has to be precisely regulated in physiological condition. For example, I3C, an AHR agonist, can reduce inflammatory response in gut, but it simultaneously accelerates formation of colonic lesion, which may become a tumor in long-term manner [Chen, Yue, et al. "Modulating AHR function offers exciting therapeutic potential in gut immunity and inflammation." Cell & Bioscience 13.1 (2023): 85.]. Therefore, it is extremely important to consider local concentration, potency, duration of action, and pharmacokinetics of AHR regulators in order to use them in clinical settings.

  1. Sample Size: The study's conclusions are based on experiments with specific AML cell lines (THP-1 and U937). Broader studies involving diverse AML samples would strengthen the generalizability of the results.

Response:

We thank for the reviewer’s comment. As the reviewer suggested, we added another AML cell line, HL-60, to confirm that our hypothesis can be further generalized. Although we could not fully perform the complete set of experiments, we confirmed that treatment of AHR activator (FICZ) increased migration rate and NMIIA expression, while AHR inhibitor (SR1) decreased both migration rate and NMIIA expression in HL-60 cells. MMP9 expression level was unchanged after treatment of FICZ or SR1 (See the figure below). These results from HL-60 matches with the results from THP-1 and U937 cells. Please excuse us that we did not test TCDD and CH223191 since we do not have leftover reagents (also, recently, Korea banned importing TCDD since it has been classified as highly toxic material). We added these results in the supplementary material section and mentioned them in the result section 2.6. (Page XX, line YY)

Revised manuscript: Result 2.6. Page XX, Line YY

To confirm that our hypothesis can be generalizable, we repeated several key experiments on HL-60 cells using the AHR agonist (FICZ) and antagonist (SR1). HL-60 cells responded well to FICZ and SR1 by significantly increasing and decreasing CYP1B1 expressions, respectively (Figure S1A). Similar to the results from THP-1 and U937 cells, HL-60 cells also showed an increased Transwell migration rate and NMIIA expression level when treated with FICZ, while the migration rate and NMIIA expression level decreased when treated with SR1, with no change in MMP9 expression (Figures S1B-S1D). 

Supplementary Figure 1

Supplementary Figure 1. (A) Relative mRNA expression levels of the CYP1B1 gene 6 and 12 hours after-dosing with FICZ and SR1. GAPDH was used for normalization. Data are shown as average ± s.e.m. One-way ANOVA is performed with Tukey’s multiple comparison tests. significance level were set as follows: *: p < 0.05, ****: p < 0.001. (B) Relative mRNA expression of MMP9 gene in HL-60 cells 12 hours after FICZ and SR1. Control is set to 1. one-way ANOVA is performed with Tukey’s multiple comparison. Significant is set to ns: non-significant. (C) Quantification of relative migration of HL-60 treated with FICZ and SR1. CTRL is set to 1 in each experiment set. (D) Western blot of NMIIA (200 kb) and β-actin (42 kDa) expressions in HL-60 cells 6 and 12 hours after treatment with FICZ and SR1.

Minor questions:

  1. Mechanism of AHR Regulation: What are the detailed molecular pathways through which AHR regulates NMIIA and other cytoskeletal components in AML cells?

Response:

Implication for NMIIA Regulation: Our findings suggest that the AHR-mediated suppression of the MYH-9 gene, which encodes NMIIA, might be a secondary outcome of the canonical AHR pathway. However, the detailed mechanisms underlying this regulation require further investigation. The interaction between AHR signaling and cytoskeletal dynamics is complex and likely involves multiple regulatory layers, including direct transcriptional control and indirect modulation through downstream signaling pathways.

Therefore, additional studies are needed to elucidate the precise molecular pathways by which AHR influences NMIIA and other cytoskeletal components in AML cells. This understanding could provide valuable insights into potential therapeutic targets for modulating cell motility and invasion in leukemia. We thank for the reviewer’s question. As aforementioned in our original manuscript, it is difficult to explain the pathway based on the conventional canonical pathway of AHR when AHR activation is inducing “downregulation” of target genes, because canonical AHR activation usually induces transcription of target gene. Therefore, we suspect that AHR activation transcribed a pool of miRNAs targeting the MYH-9 mRNA. There is a report that AHR-mediated over expression of miR-212/132 cluster reduced migration and invasion of breast cancer cells by suppressing mRNA of pro-metastatic transcription factor SOX-4 [Disner, Geonildo Rodrigo, Monica Lopes-Ferreira, and Carla Lima. "Where the aryl hydrocarbon receptor meets the microRNAs: literature review of the last 10 years." Frontiers in Molecular Biosciences 8 (2021): 725044.]. We discussed this issue in our discussion section.

Revised manuscript: Discussion / Page XX, Line YY

However, our findings, which show AHR-mediated suppression of the MYH-9 gene, suggest that this effect might be a secondary outcome of the canonical AHR pathway. For example, AHR can induce the transcription of microRNAs (miRNAs) to downregulate the transcription of target mRNAs. Therefore, we suspect that AHR activation transcribed a pool of miRNA targeting the MYH-9 mRNA. There is a report that AHR-mediated overexpression of miR-212/132 cluster reduced migration and invasion of breast cancer cells by suppressing the mRNA of the pro-metastatic transcription factor SOX-4 [ref ]. Further investigation into detail mechanism is required.

  1. Clinical Feasibility: Can AHR agonists be safely and effectively used in patients, and what would be the potential side effects?

Response:

Thank you for bringing up this significant point. We have elaborated that the lowered toxicity of naturally derived AHR ligands compared to exogenous synthetic AHR ligands such as TCDD. As the reviewer pointed out, AHR regulators are double-sided sword, which can exert both positive and negative effects. For an example, I3C, an AHR agonist, can reduce inflammatory response in gut, but it simultaneously accelerates formation of colonic lesion, which may become a tumor in long-term manner [Chen, Yue, et al. "Modulating AHR function offers exciting therapeutic potential in gut immunity and inflammation." Cell & Bioscience 13.1 (2023): 85.]. Therefore, it is extremely important to consider local concentration, potency, duration of action, and pharmacokinetics of AHR regulators in order to use them in clinical settings. We added aforementioned statements in our discussion section.

Revised manuscript: Discussion / Page XX, Line YY

Given these concerns, there is an urgent need to identify AHR agonists that are safe and effective for systemic use. Recently, naturally derived compounds like indole-3-carbinol, resveratrol, curcumin, and polyphenols have been proposed as alternatives to conventional AHR agonists, exhibiting relatively lower toxicity than exogenous synthetic AHR ligands such as TCDD [27]. However, use of AHR agonist or antagonist in clinical setting should be cautious since the level of AHR activation has to be precisely regulated in physiological condition. For example, I3C, an AHR agonist, can reduce the inflammatory response in the gut, but it simultaneously accelerates the formation of a colonic lesion, which may become a tumor in a long-term manner [Chen, Yue, et al., "Modulating AHR function offers exciting therapeutic potential in gut immunity and inflammation." Cell & Bioscience 13.1 (2023): 85.]. Therefore, it is extremely important to consider local concentration, potency, duration of action, and pharmacokinetics of AHR regulators in order to use them in clinical settings.

  1. Broader Implications: How does AHR activity interact with other signaling pathways in AML and other types of leukemia?

Response:

We thank for the reviewer’s comment. As the reviewer suggested, chemokine receptors of AML cells such as CXCR4 or CCR5 can affect their migration and invasion [Faaij, Claudia MJM, et al., "Chemokine/chemokine receptor interactions in extramedullary leukaemia of the skin in childhood AML: differential roles for CCR2, CCR5, CXCR4 and CXCR7." Pediatric blood & cancer 55.2 (2010): 344–348] We discussed this issue in our discussion part as the limitation of this study.

Revised manuscript: Discussion/Page XX, Line YY

“Another limitation of this study is that only MMP9 and NMIIA were examined as down-stream targets of AHR. Expressional changes of chemokine receptors such as CXCR4 or CCR5 can also affect migration and invasion of AML cells [34]. Therefore, it should also be further examined whether activation or inhibition of AHR can alter expression levels of those genes.”

  1. Long-term Effects: What are the long-term outcomes of modulating AHR activity in AML treatment? Will it lead to sustained remission or just a temporary reduction in invasiveness?

Response:

Thank you for mentioning this point. Although there is no direct evidence, we believe that the treatment of AHR activator will only exert a temporary effect since the clinically applicable AHR activators, including natural AHR ligands, have limited half-life ranged between 12 and 24 hours [Busbee, Philip B., et al., "Use of natural AhR ligands as potential therapeutic modalities against inflammatory disorders." Nutrition reviews 71.6 (2013): 353–369.]. After that, they are completely metabolized. However, this limited time of action can also be beneficial, since long-term activation of AHR may induce tumorigenesis of other organs [Chen, Yue, et al., "Modulating AHR function offers exciting therapeutic potential in gut immunity and inflammation." Cell & Bioscience 13.1 (2023): 85.]. Therefore, in the clinical setting, there should be significant considerations given to the local concentration, potency, duration of action, and pharmacokinetics of AHR regulators. We discussed this issue in the discussion section.

Revised manuscript:  Discussion / Page XX, Line YY

Given these concerns, there is an urgent need to identify AHR agonists that are safe and effective for systemic use. Recently, naturally derived compounds like indole-3-carbinol, resveratrol, curcumin, and polyphenols have been proposed as alternatives to conventional AHR agonists, exhibiting relatively lower toxicity than exogenous synthetic AHR ligands such as TCDD [27]. However, use of AHR agonist or antagonist in clinical setting should be cautious since the level of AHR activation has to be precisely regulated in physiological condition. For example, I3C, an AHR agonist, can reduce inflammatory response in gut, but it simultaneously accelerates formation of colonic lesion, which may become a tumor in long-term manner [ref ]. Therefore, it is extremely important to consider local concentration, potency, duration of action, and pharmacokinetics of AHR regulators in order to use them in clinical settings.
